# On the Use of Waste Materials for Thermal Improvement of 3D-Printed Block—An Experimental Comparison

Tullio de Rubeis *, Annamaria Ciccozzi, Giovanni Pasqualoni, Domenica Paoletti and Dario Ambrosini

Department of Industrial and Information Engineering and Economics, University of L'Aquila,
67100 L'Aquila, Italy; annamaria.ciccozzi@graduate.univaq.it (A.C.); giovanni.pasqualoni@univaq.it (G.P.);
domenica.paoletti@univaq.it (D.P.); dario.ambrosini@univaq.it (D.A.)
* Correspondence: tullio.derubeis@univaq.it

**Abstract:** Over the years, the building envelope has evolved from a protective barrier element to a complex filter system capable of optimizing the interactions between the external and internal environments. An efficient envelope reacts flexibly to variable external conditions, minimizing heat losses in the winter season. Therefore, insulating materials play a fundamental role in building's thermal performance. In this scenario, Additive Manufacturing represents an emerging and promising solution for the construction sector. Three-dimensional printing allows the creation of custom geometries, reduces material waste, and automates the construction process. This work aims to compare the thermal performance of a PLA (polylactic acid) 3D-printed block with an internal honeycomb structure whose air cavities are filled with natural and recyclable waste-insulating materials. The selected air cavity filling materials are (i) wood sawdust, (ii) sheep's wool, and (iii) hemp. The thermal behavior of the block with the different filling materials was experimentally tested via Heat Flow Meter (HFM) method in a controlled environment (Hot Box). The results showed that the introduction of waste material significantly improved the thermal performance of the 3D-printed block compared to the case of air cavities. A thermal transmittance (*U*-value) reduction of up to 57% was obtained. Moreover, the sheep's wool showed the best performance, with a *U*-value equal to $0.53 \pm 0.02$ W/m$^2$K, i.e., 18.5% less than the wood sawdust and 19.7% less than hemp.

**Keywords:** thermal insulation; insulating materials; air cavities; 3D printing; Hot Box analysis; infrared thermography; heat flux meter; energy efficiency; sustainability





## 1. Introduction

In 2021, buildings accounted for 30% of global final energy consumption and 27% of GreenHouse Gas (GHG) emissions (8% of which are direct emissions, while 19% occur due to electricity and heat production) [1]. Only in 2019, energy consumption and $CO_2$ have reduced due to the COVID-19 pandemic [1]. The performance requirements along with European energy directives are speeding up the spread of renewable and highly efficient technologies [2]. At the same time, efforts to determine new energy-efficient solutions are increasing to reach the Net Zero Emissions objective by 2050. Therefore, the next decade is crucial to implement the necessary energy efficiency measures in the building sector [1].

The building energy consumption strongly depends on the characteristics of its envelope that influence the transmission losses between the heated volume and the external environment. Therefore, thermal insulation is undoubtedly one of the best ways to reduce energy consumption. Insulating materials play a key role in this scenario as choosing the correct material, its thickness, and its position allows for good indoor thermal comfort conditions and adequate energy savings [3,4]. The thermal properties of insulating materials are extremely important; however, they are not the only aspects to be considered [5]. From a sustainability perspective, the assessment of the greenhouse effect of the building components is becoming increasingly important [6–11].

To date, building insulation is commonly achieved using materials obtained from petrochemicals (mainly polystyrene) or natural sources processed with high energy consumption (e.g., glass and rock wool). These materials cause significant harmful environmental effects mainly during the production (due to the high consumption of fossil fuels) and disposal phases (due to problems of reuse or recycling at the end-of-life). The introduction of the "sustainability" concept in the building design process has encouraged research aimed at the use of natural or recycled thermal and acoustic insulating materials.

The insulating materials of synthetic origin (e.g., polystyrene, polyester, polyethylene, etc.) are thermally very high-performing, but they present some critical issues. They are non-renewable, emit gas harmful to human health, have high GHG effects, and are hardly recyclable. On the contrary, materials of natural origin have a low carbon footprint, especially when they are produced near the construction site, limiting the GHG emissions for transportation. Among the materials of natural origin, hemp fiber, cellulose fiber, wood fiber, mineralized wood wool, expanded cork, and sheep's wool are some of the most promising.

Füchsl et al. [12] presented a systematic literature review on thermal insulation materials. Their study showed that the most studied renewable materials are cork [13–22], cellulose [13,14,17,20,23–26], and hemp [17,19,24,25,27–30].

Islam et al. [31] provided a realistic picture of the current state and potential for the use of recycled textiles as insulation materials. In their study, the environmental effects of textile waste and its conversion into insulating building materials are discussed.

Bisegna et al. [32] discussed the effects of different configurations of the envelope (expanded polystyrene, glass wool, wood fiber, and kenaf) on an Italian residential building on both energy and environmental performance. The evaluation was performed through two different green building assessment methods, such as LEED and the Italian ITACA.

Briga-Sa et al. [33] studied the potential applicability of woven fabric waste (WFW) and a waste of this residue, named woven fabric subwaste (WFS), as thermal insulation building material. Experimental work was conducted using an external double wall, with the air-box filled with these two types of waste, to determine their thermal characteristics. Two heat flow meters and four surface temperature sensors were placed on the wall surface to determine the thermal conductivity of the wastes. The obtained results show that the application of the WFW and WFS in the external double wall increases its thermal behavior by 56% and 30%, respectively. The thermal conductivity value of the WFW is like the values obtained for expanded polystyrene (EPS), extruded polystyrene (XPS), and mineral wool (MW).

Zach et al. [34] reported that sheep wool has excellent thermal and acoustic insulation properties and comparable characteristics with mineral/stone wool. In addition, sheep's wool is more environmentally friendly and less harmful to health than mineral wool.

Asdrubali et al. [35] presented an updated review of some products for the insulation of buildings made with unconventional materials, deriving from natural sources such as residues from agricultural production and processing industries. Choosing sugar cane, pineapple, and rice residues as insulation materials could help reduce the use of petroleum and non-renewable sources.

In this scenario, 3D printing (3DP) has high potential due to its capabilities for creating complex geometries, automating construction processes, and minimizing waste production. The great potential of Additive Manufacturing (AM) also makes it possible to create thermal insulation blocks that can be filled with waste materials. Therefore, two of the main reasons why 3DP can be considered a promising technology to produce insulating blocks are:

(1)　the topological optimization of the structure, creating complex geometries suitable for mitigating heat transfer phenomena;

(2)　the possibility of using the 3D-printed blocks as housing for waste insulation materials.

However, the interaction between 3DP and the use of waste materials, potentially useful for the thermal insulation of buildings, is very poorly discussed, even though in recent years, the use of 3DP for thermal purposes has attracted growing interest among researchers.

Grabowska and Kasperski [36] designed and printed multi-layer materials, with quadrangle, hexagonal, and triangle closures to individuate structures with lower thermal conductivity.

Distinct printable wall configurations with different materials were designed by Alkhalidi et al. [37] to reduce the *U*-values of the painted walls in compliance with climatic zone regulations.

Suntharalingam et al. [38] studied the thermal transmittance value (*U*-value) of 32 different 3DPC wall configurations with and without cavity insulation using validated finite element models. The goal of the study was to identify the best-performing geometric configurations at a thermal level.

The sustainable aspects of 3DP, such as the possibility of using waste material, minimization of the supply chain and emissions, less time spent on post-processing, and reduced costs for making complex products make it a technology of the future [39]. Precisely for this reason, it is necessary to explore the potential of this emerging technology. It could be the key solution capable of positively influencing the energy and environmental footprint of buildings [40].

Based on the above, it could be interesting to integrate the advantages offered by 3D printing with those offered by insulating materials with a low carbon footprint. The research proposed in the article could be the starting point for future developments in the field of energy saving in buildings.

This work intends to continue previous research [41,42] on the analysis of 3DP potential for the thermal performance optimization of the building envelope. de Rubeis theoretically and experimentally analyzed the thermal performance of a 3D-printed PLA block [41]. The PLA was chosen because it is one of the most widely used materials for 3DP and is a recyclable material with low GHG emissions [43]. PLA is a bioplastic of natural origin derived from sugars obtained from corn which; for some years, it has been used in the production of fibrous insulation. The study was initially conducted with empty air cavities of the 3D-printed block, then the air cavities were filled with waste materials such as polystyrene and wool.

Later, de Rubeis et al. [42] theoretically and experimentally analyzed the thermal behavior of three PLA 3D-printed blocks characterized by different internal geometries. Multi-row, square, and honeycomb structures were designed and investigated. The results showed that as the complexity of the blocks' internal structure increased, a heat flow reduction could be observed. In fact, the honeycomb structure showed better behavior than the other two blocks, with an experimental transmittance value equal to $1.22 \pm 0.04 \text{ W/m}^2\text{K}$.

Therefore, this work aims to expand the previous studies [41,42] to understand how the re-use of waste materials, such as wood sawdust, sheep's wool, and hemp, can improve the thermal behavior of a PLA 3D-printed block by filling its air cavities. Specifically, based on the previous works [41,42] in which different internal geometries of 3D-printed blocks were compared, the block with the best thermal properties was selected.

## 2. Materials and Methods

Thermal insulating materials play a fundamental role in reducing buildings' energy consumption due to their thermal characteristics that minimize heat transmission losses [44]. Good thermal insulation could save about 65% of energy consumption in domestic buildings [45].

Many natural and waste materials can be used for the thermal insulation of building envelopes [3,5–30,46–53]. However, the choice made in this work was based on the availability and the possibility of using materials that can be employed to fill the air cavities of the 3D-printed blocks [38,41,54,55].

Therefore, a comparison of the thermal performance of three natural and waste materials, i.e., wood sawdust, sheep's wool, and hemp, filled in a 3D-printed PLA block with an internal honeycomb structure is presented.

### 2.1. Methodology

The objective of this work is to integrate the potential of 3DP technology with the advantages offered by the reuse of waste-insulating materials, with interesting thermo-physical properties such as air cavity filling materials of a PLA 3D-printed block with a honeycomb internal structure. The analyses were conducted experimentally, and the results were compared with those obtained in the previous study [42] in terms of thermal transmittance values.

The methodology used in this study can be divided into two macro-phases, as shown in Figure 1.

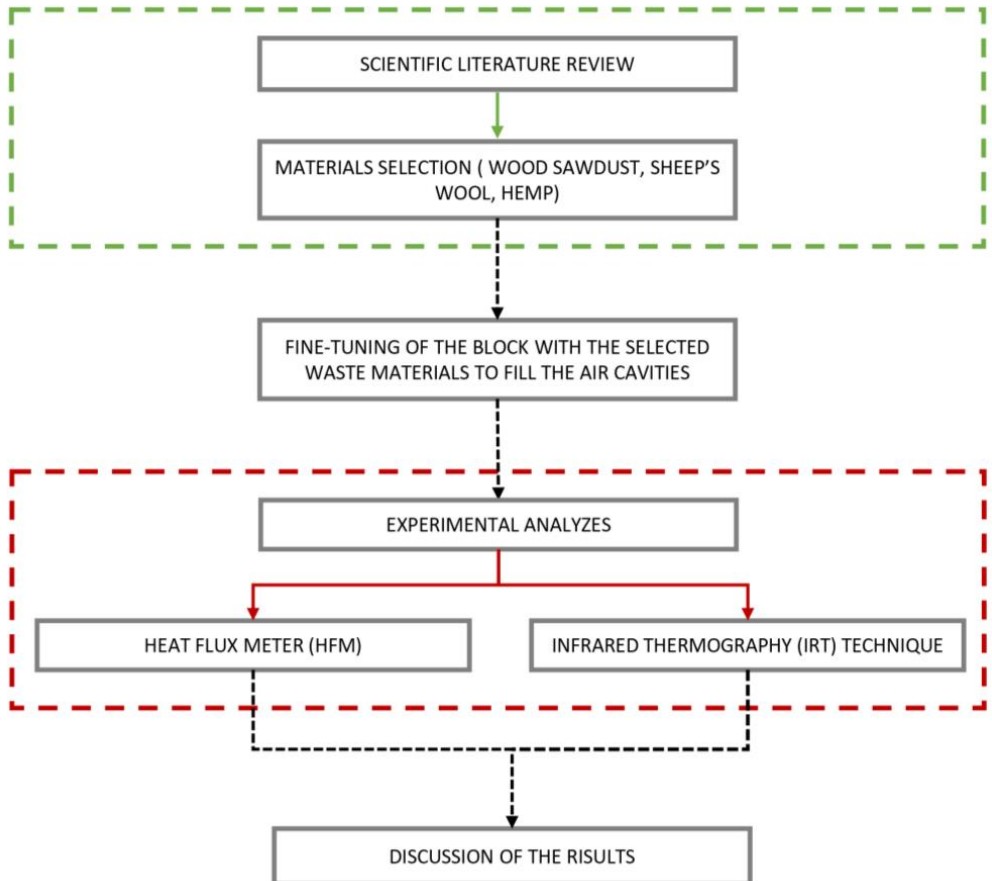

**Figure 1.** Methodology flowchart representing the various work phases.

The first phase focused on the choice of insulating materials to be inserted into the cavities of the block:

- research on the various insulating materials;
- choice of three insulating reuse materials [3,5–30,46–53].

The second phase focused on the study of the block's thermal behavior with the cavities filled with the chosen insulating materials:

- experimental analyses in Hot Box were conducted using Heat Flux Meter (HFM);
- then, a survey was performed through InfraRed Thermography (IRT) technique.

### 2.2. The 3D-Printed Block

The block used for the experimental analysis has a dimension of 250 × 250 × 100 mm (width × height × depth) and it is characterized by an internal honeycomb configuration formed by hexagonal cavities. Figure 2 shows block drawings made in AutoCAD® 2D and 3D design software.

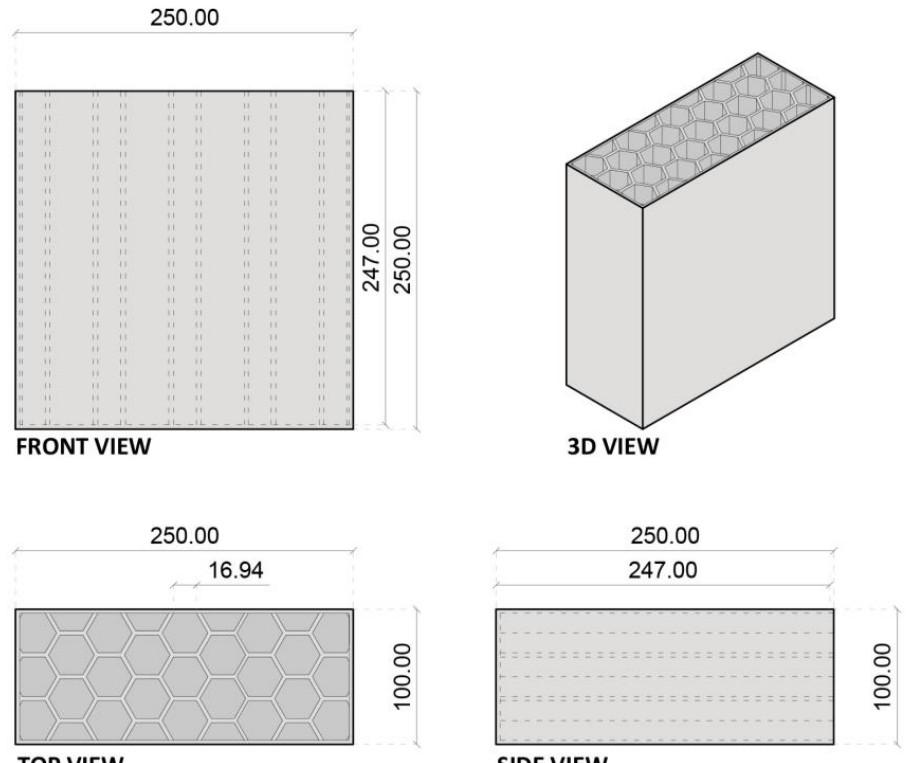

**Figure 2.** 2D and 3D block design (measurements in millimeters). Remark: PLA thickness is equal to 3 mm.

The creation of the block occurred in three steps. The first step consisted of three-dimensional modeling, conducted with the use of AutoCAD Inventor®.

Then, the 3D model of the block was inserted in the Creality Slicer 4.2 slicing software for the assignment of all the printing parameters. The main printing parameters were the characteristics of the shell and the filling, the printing material and the temperature of the nozzle and print bed, the print speed, and the nozzle size. The block was printed with a brass nozzle (diameter of 0.4 mm), and the printing temperature was set at 210 °C, suitable for PLA. The temperature of print bed temperature was placed at 60 °C. Following the assignment of the various parameters, the software generated the various print layers, necessary for the creation of the G-Code file.

Finally, the block was printed with the Creality CR-3040 PRO 3D printer.

**Remark 1.** *Design, modeling, and printing phase are detailed in* [42].

### 2.3. Selection of Insulating Materials

After a thorough search in the literature [3,5–30,46–53], three waste-insulating materials were chosen for the study. Starting from the literature study, the choice was made based on the availability of the materials and the ease of inserting materials into the air cavities of the 3D-printed block with a honeycomb structure.

The wood sawdust (Figure 3a) is an insulating material of vegetable origin. The wood fiber used as thermal insulation can be considered a sustainable material if it is derived from sustainable forestry or from processing waste (e.g., carpentry or forest maintenance operations). The thermal conductivity of this material ranges from 0.038 to 0.050 W/mK, its specific heat varies between 1.9 and 2.1 kJ/kgK, and its water vapor diffusion resistance factor (μ-value) ranges from 1.0 to 5.0 [3]. In this study, the properties of wood waste from primary production sources using untreated material were examined. These residues are industrial wastes generated by carpentry or other wood processing companies. The use of this material without the addition of binders facilitates better management of wood waste,

ease of recycling, and potentially healthier indoor environments. Wood fibers are currently used in the production of wood fiber insulation boards by adding small amounts of PUR resin (Polyurethane Resin) in a dry process. However, this manufacturing process also requires a large amount of energy. The use of wood waste received from local carpentry without treatment reduces energy consumption and the consequent release of carbon dioxide [56]. The second material selected in this work is sheep's wool (Figure 3b). This material, virgin or recycled, has the highest ozone depletion potential. It is renewable, recyclable, and ecological. Furthermore, sheep's wool has excellent thermal and acoustic insulation properties [50]. Its thermal conductivity varies between 0.038 and 0.054 W/mK, its specific heat ranges from 1.3 to 1.7 kJ/kgK, its water vapor diffusion resistance factor ($\mu$-value) is between 1.0 and 3.0 [3]. The last selected material is hemp (Figure 3c), which is an insulating material of vegetable origin deriving from ecological, recyclable, and renewable cultivation. Hemp is increasingly expanding its field of application in the construction industry [57–61]. What makes hemp highly suitable for building applications is mainly its good hygrothermal power and its soundproofing properties. Moreover, natural fibers have a very high carbon storage potential [30]. Due to the biogenic uptake of $CO_2$ during hemp growth, hemp has a negative carbon footprint and thus acts as an effective carbon sink. The net carbon sequestration by the industrial hemp crop is estimated at 0.67 tons/ha (hectare)/year [62]. The thermal conductivity of this material ranges between 0.038 and 0.060 W/mK, its specific heat is between 1.9 and 2.1 kJ/kg K and its water vapor diffusion resistance factor ($\mu$-value) ranges from 1.0 to 2.0 [3].

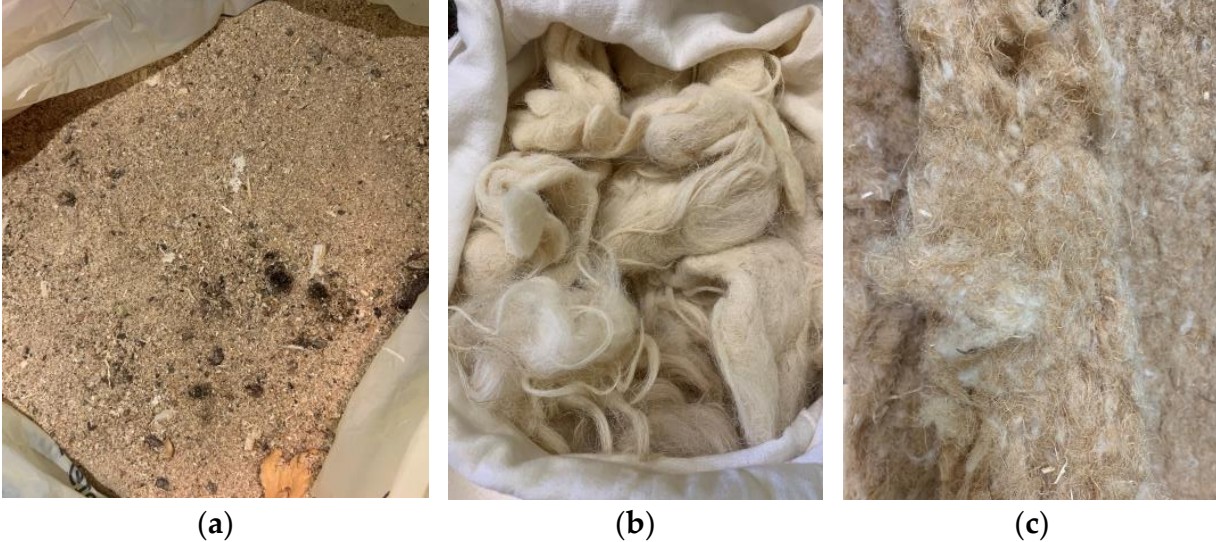

| (a) | (b) | (c) |

**Figure 3.** Materials chosen for the study. (**a**) Wood sawdust. (**b**) Sheep's wool. (**c**) Hemp.

The choice fell on natural or recycled insulating materials because, unlike those of synthetic origin (polystyrene, polyester, polyethylene, etc.), they have a much lower carbon footprint. Frequently, natural insulating materials are used with additives to prevent decomposition phenomena and ensure fire safety [12]. However, additives lead to an increased carbon footprint of natural insulating materials. In the present work, the chosen insulating materials were not treated with the addition of additives.

Three samples were set up to conduct the experimental analysis. The first sample was filled with 1107.9 g of wood sawdust, the second with 339 g of sheep's wool, and the third with 427.3 g of hemp. Considering that the weight of the empty block is 571 g, the density of the filler wood sawdust is 237.32 kg/m$^3$, that of sheep's wool is 72.62 kg/m$^3$ and that of hemp is 91.53 kg/m$^3$. Figure 4 shows the block placed on the scale with the three different filling materials.

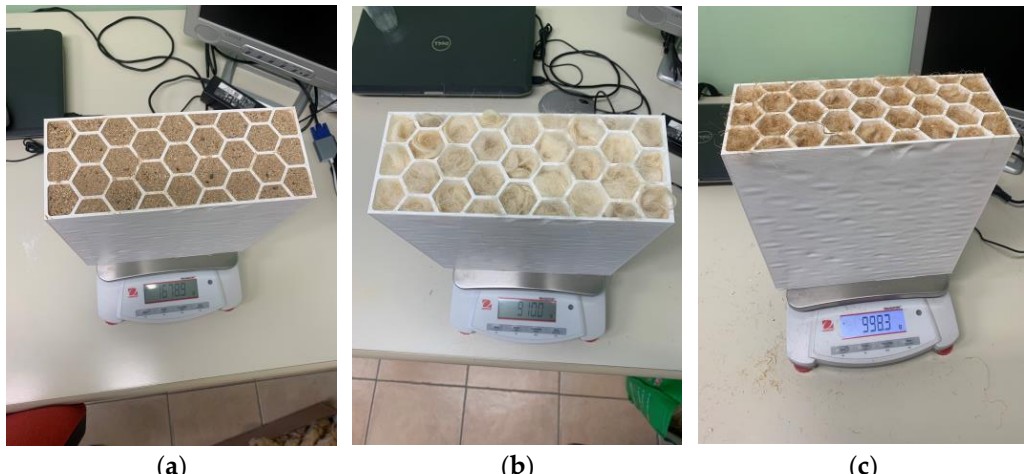

|   |   |   |
|---|---|---|
| (**a**) | (**b**) | (**c**) |

**Figure 4.** Weight of the three samples. (**a**) Block with wood sawdust. (**b**) Block with sheep's wool. (**c**) Block with hemp.

### 2.4. Analysis Phase

The experimental analyzes were conducted using the heat flux meter (HFM) method and infrared thermography (IRT) technique.

The heat flux measurements were performed through an especially built Hot Box [42,63,64]. The dimensions of the Hot Box are 640 × 340 × 360 mm (length × height × width). It is composed of a hot chamber, where an electric heater brings the temperature to the established value, a baffle inserted near the inner face of the 3D block to separate the hot chamber from the laboratory environment, and walls realized with insulating material, covered with sheet steel [41].

The analyses were carried out in the laboratory, at an average air temperature of about 22 °C. For this reason, the temperature of the hot chamber was set to 54 °C to have an adequate thermal gradient between the two surfaces of the block as provided by the ISO 9869 standard [65]. The measurements were performed with two surface temperature probes placed on both surfaces of the block, one air temperature sensor inside the Hot Box, and the heat flux sensor installed on the internal side of the block (Figure 5). Each analysis lasted 24 h, during which the data logger acquired data every 10 min.

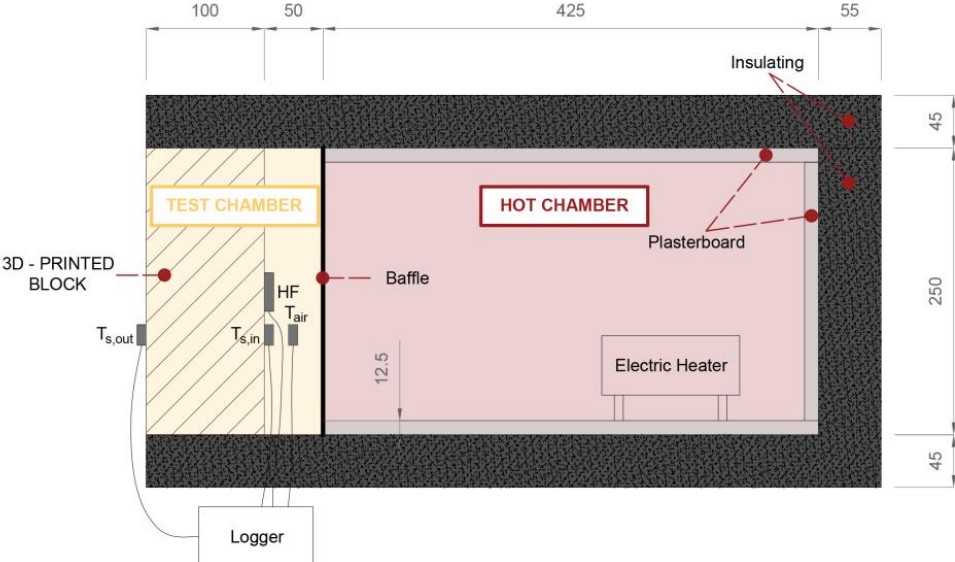

**Figure 5.** Configuration used for the HFM analyses.

As indicated by the ISO 9869 standard [65], the progressive average method was employed to obtain conductance ($\Lambda$) and transmittance ($U$) values, using the Equations (1) and (2), respectively:

$$\Lambda = \frac{\sum_{j=1}^{n} q_j}{\sum_{j=1}^{n} \left(T_{s,in,j} - T_{s,out,j}\right)} \ [\text{W/m}^2\text{K}], \tag{1}$$

$$U = \frac{1}{R_{tot}} \ [\text{W/m}^2\text{K}], \tag{2}$$

where

- $\sum_{j=1}^{n} \left(T_{s,in,j} - T_{s,out,j}\right)$ is the progressive sum of the differences between internal and external surface temperatures;
- $\sum_{j=1}^{n} q_j$ is the progressive sum of the density of the heat flux;
- $R_{tot}$ is the total thermal resistance that also includes the internal ($R_{s,i}$) and external ($R_{s,e}$) thermal resistances taken from the EN ISO 6946 standard [66] equal to 0.13 m$^2$K/W and 0.04 m$^2$K/W, respectively.

After each HFM experimental campaign, when the surfaces of the block reached stationary thermal conditions, the IRT technique was performed using a FLIR ThermaCAM® T1020 IR camera (TELEDYNE FLIR, Milan, Italy), at 1.50 m from the front of the block and at a distance of 0.50 m from the top, following the scheme shown in Figure 6 [42].

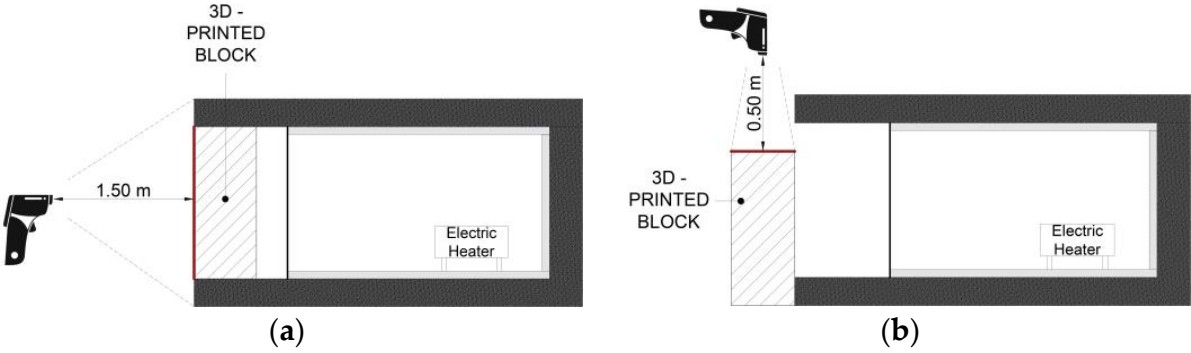

**Figure 6.** Configuration used for the IRT technique. (**a**) IRT technique in front of the block. (**b**) IRT technique from the top of the block.

The experimental analyzes were conducted using the measuring instruments shown in Table 1 [42].

**Table 1.** Details of the measuring instruments used for the experimental analyses.

| Sensor | Type | Measuring Range | Resolution |
|---|---|---|---|
| Heat flow meter | Hukseflux HFP01 | From −2000 to 2000 W/m$^2$ | $60 \times 10^{-6}$ V/(W/m$^2$) |
| Surface temperature | LSI Lastem DLE 124 | From −40 to 80 °C | 0.01 °C |
| Air Temperature | LSI Lastem DLA 033 | From −40 to 80 °C | 0.01 °C |
| Datalogger | LSI Lastem M-Log ELO008 | From −300 to 1200 mV | 40 µV |
| IR camera | FLIR T1020 | From −40 to 2000 °C | 1024 × 728 pixel |

## 3. Results

The experimental campaigns had a total duration of 72 h. The first analysis was conducted on the block with the cavities filled with wood sawdust. In this case, the measurements started on 10 February 2023 at 11:30 a.m. and ended on 11 February at 11:30 a.m. (Figure 7a). The second cycle of measurements was carried out on the block filled with sheep's wool from 16 February at 7:30 p.m. until 17 February at 7:30 p.m.

(Figure 7b). Finally, from 27 February at 19:10 a.m. to 28 February at 19:10 a.m., the analyses were conducted on the last sample, i.e., the block with the cavities filled with hemp (Figure 7c).

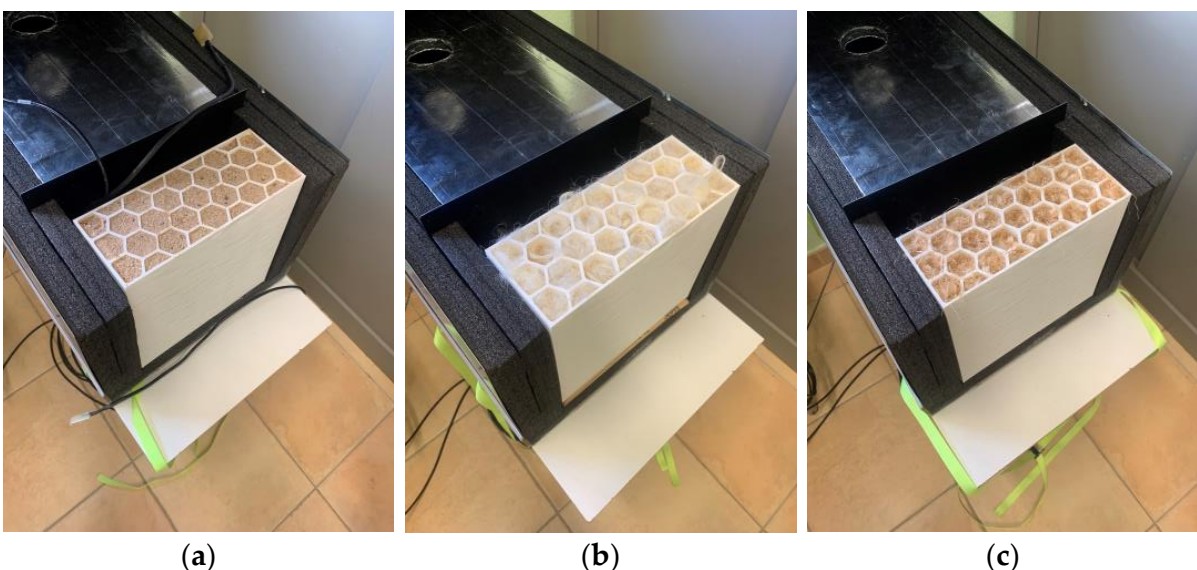

**Figure 7.** The three samples placed inside the Hot Box. (**a**) Block with wood sawdust. (**b**) Block with sheep's wool. (**c**) Block with hemp.

The conductance ($\Lambda$) and transmittance ($U$) values obtained for each sample are summarized in Table 2. The Holman's method was employed to carry out the uncertainty analysis and propagation of uncertainty [67].

**Table 2.** Experimental results (values in W/m$^2$K).

| Sample | $\Lambda$ | $U$ |
|---|---|---|
| Block with wood sawdust | $0.74 \pm 0.02$ | $0.65 \pm 0.02$ |
| Block with sheep's wool | $0.58 \pm 0.02$ | $0.53 \pm 0.02$ |
| Block with hemp | $0.74 \pm 0.02$ | $0.66 \pm 0.02$ |

The graphs in Figure 8 compare the conductance and transmittance values of the three samples.

The block filled with sheep's wool is the one that showed the best thermal behavior among the three blocks. The transmittance value resulted in 18.5% lower than that of wood sawdust and 19.7% lower than that of hemp. It is worth noting that the thermal behavior of the block filled with wood sawdust and hemp was remarkably similar.

The IRT technique, performed after each HFM analysis, helped to highlight the thermal stratifications of each sample. The thermal images in Figure 9 show the vertical thermal stratification, while those in Figure 10 show the thermal stratification from a top view.

The thermal images highlight that the block filled with wood sawdust has less vertical thermal stratification and more homogeneous thermal distribution. These results are due to a better distribution of the insulating material inside the cavities.

Finally, the results were then compared with those obtained in the previous study [42] where the analyzes were conducted with empty block cavities (Table 3).

The comparison shows that the thermal performance of the block improved significantly due to the inclusion of the chosen insulating materials in the cavities. With the introduction of sheep's wool into the block, the transmittance decreased by more than 50%.

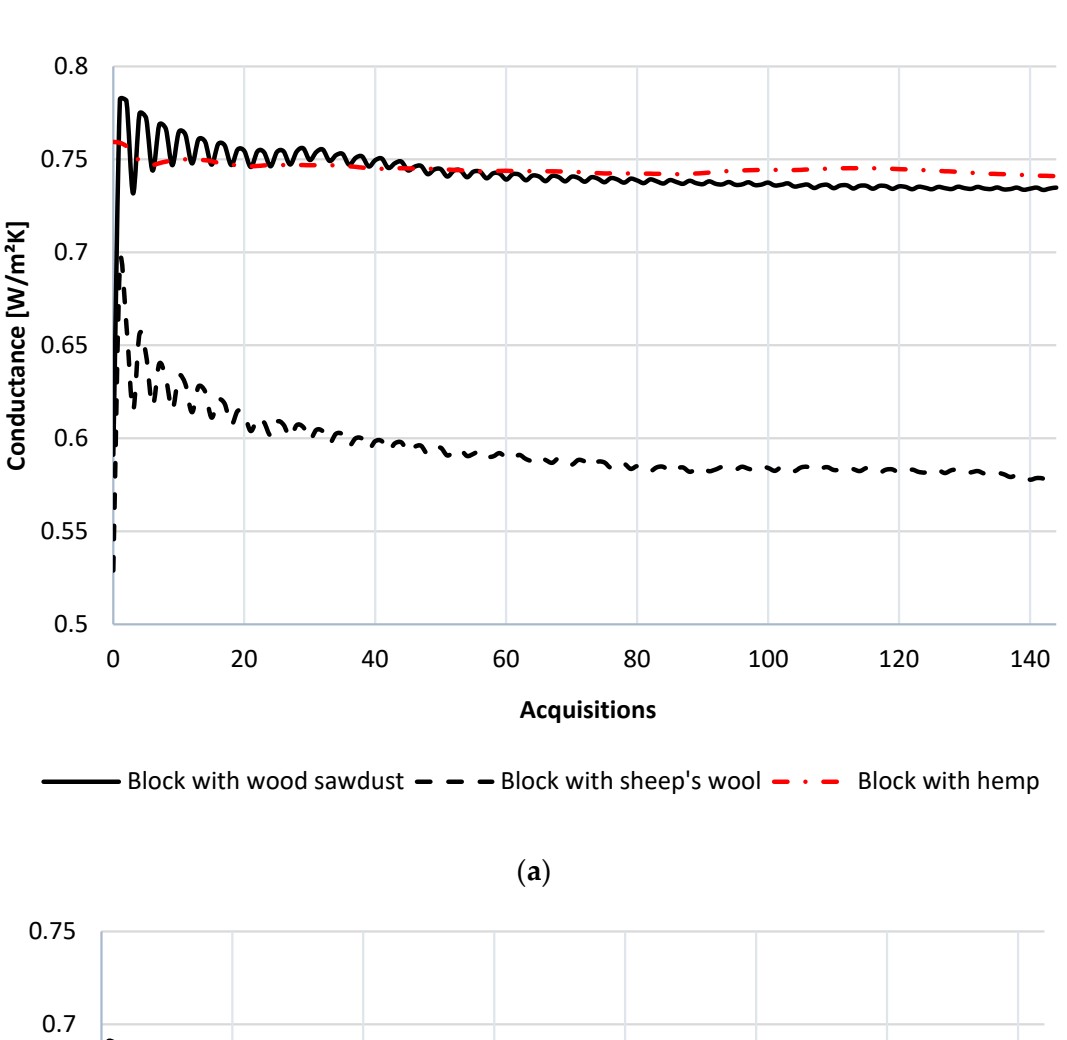

(**a**)

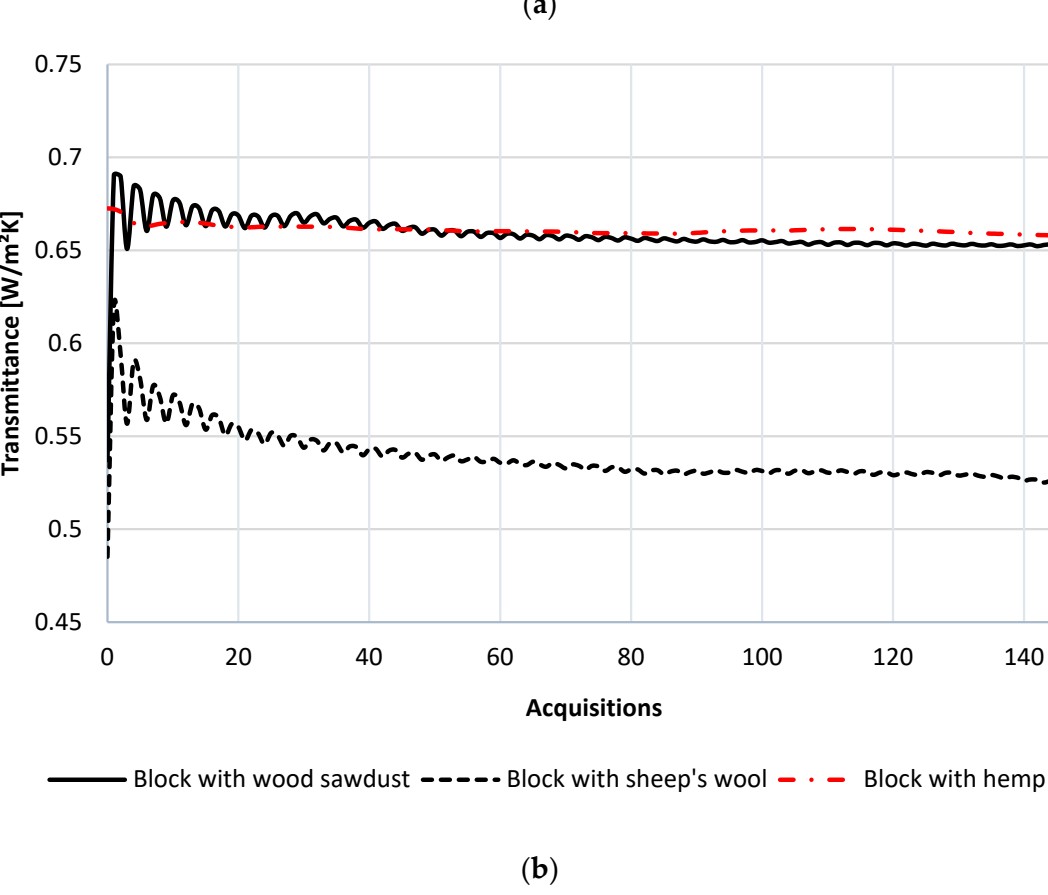

(**b**)

**Figure 8.** Conductance (**a**) and transmittance (**b**) experimental results.

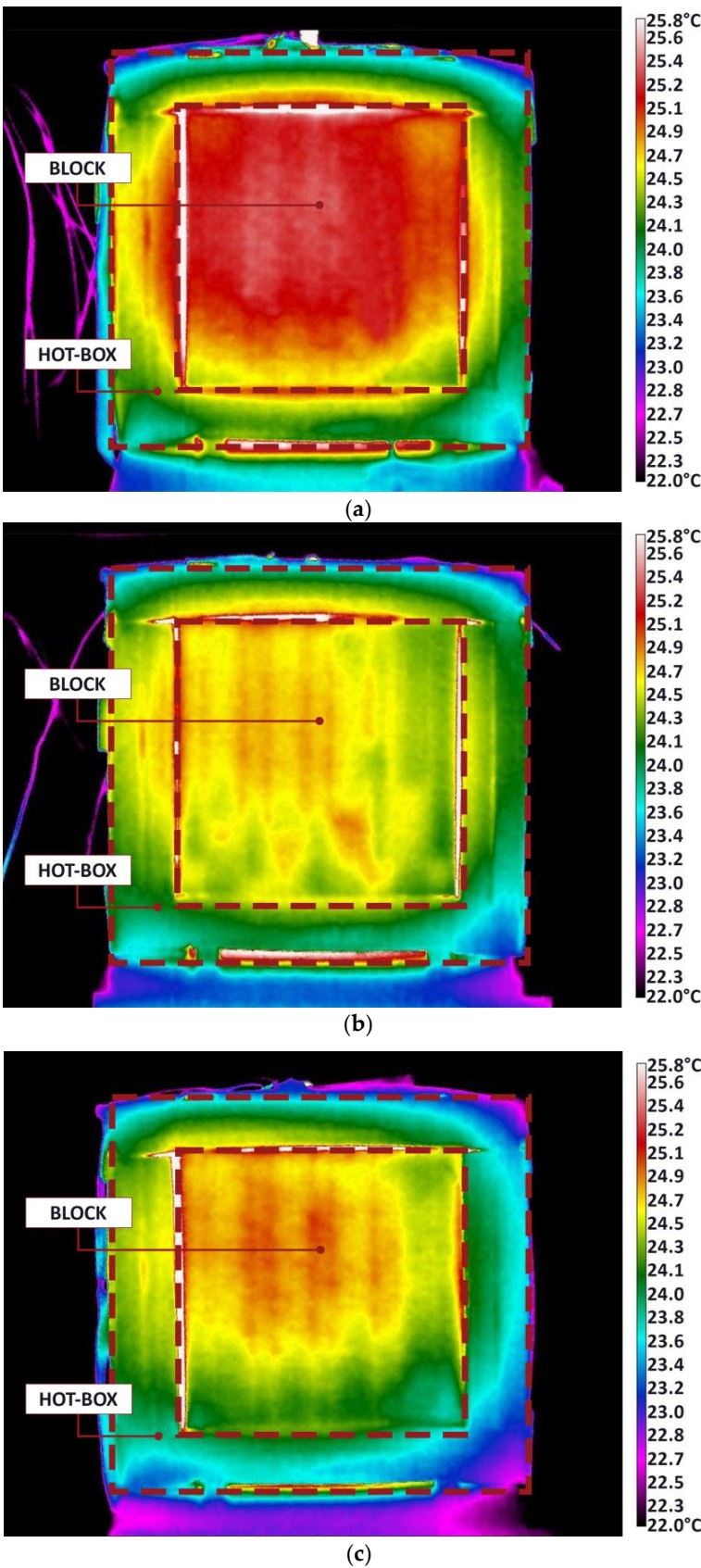

**Figure 9.** Thermal images—front view. (**a**) Block with wood sawdust. (**b**) Block with sheep's wool. (**c**) Block with hemp.



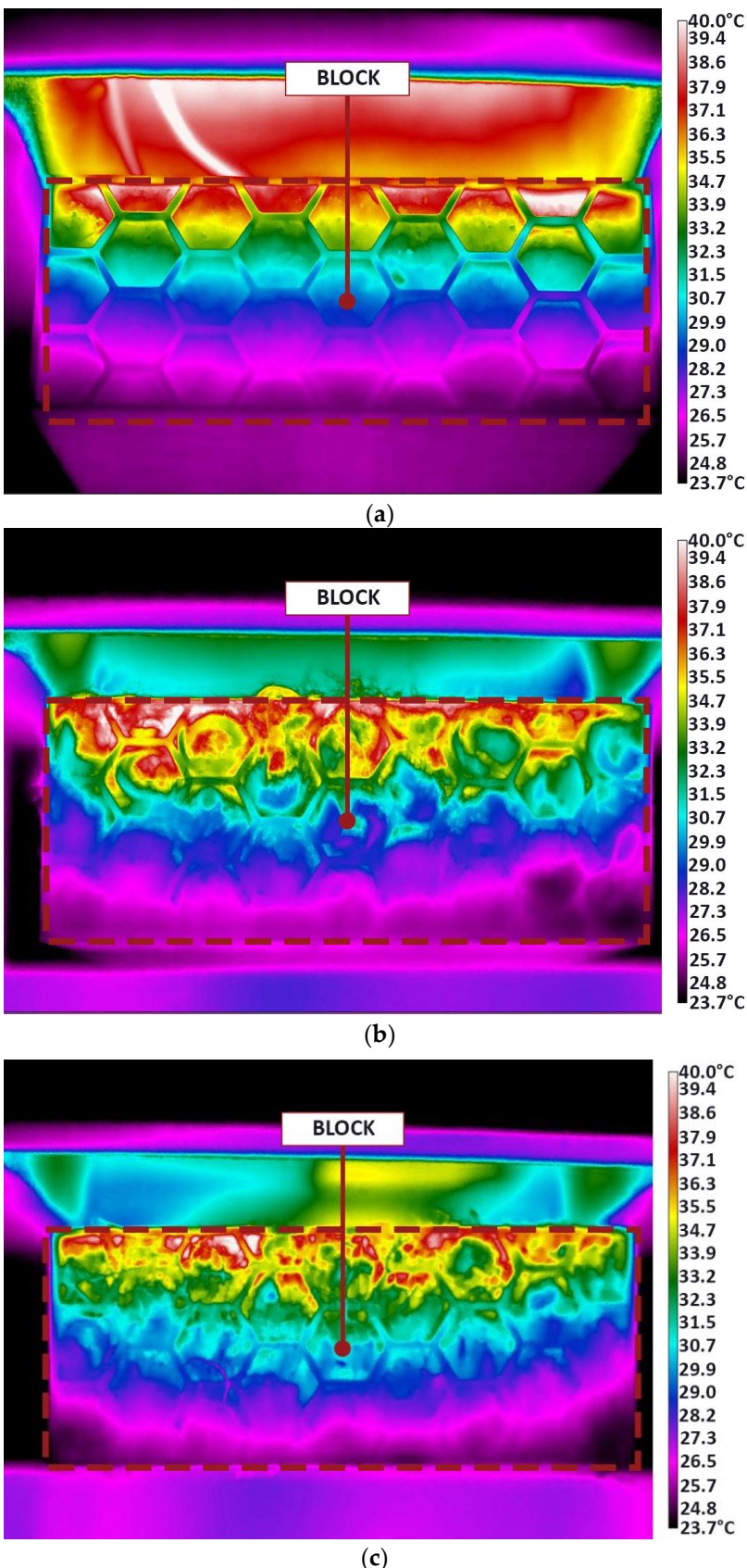

**Figure 10.** Thermal images—top view. (**a**) Block with wood sawdust. (**b**) Block with sheep's wool. (**c**) Block with hemp.

**Table 3.** Comparison of the transmittance values of the empty block and the block filled with waste-insulating materials.

| Block Type | *U*-Value [W/m$^2$K] | Percentage Difference [%] |
|---|---|---|
| Empty block | $1.22 \pm 0.04$ | 0 |
| Block with wood sawdust | $0.65 \pm 0.02$ | $-46.7$ |
| Block with sheep's wool | $0.53 \pm 0.02$ | $-56.6$ |
| Block with hemp | $0.66 \pm 0.02$ | $-45.9$ |

## 4. Conclusions

Many parameters should be taken into consideration when selecting thermal insulation, including durability, cost, compressive strength, water vapor absorption and transmission, fire resistance, ease of application, carbon footprint, and thermal conductivity. However, the thermal resistance of insulating materials is one of the main properties to be considered for building energy savings.

The adoption of 3DP in the construction sector can be a promising technology providing innovative solutions, including the possibility of making complex and optimized products, better time management, faster production, costs and waste materials reduction, and a lower greenhouse effect.

In this work, the advantages offered by 3DP have been integrated with those offered using waste natural or recycled materials. In detail, the thermal performances of a PLA 3D-printed block, characterized by an internal honeycomb structure, were analyzed. The cavities were filled with several types of waste-insulating materials: wood sawdust, sheep's wool, and hemp. The results were then compared with those obtained in a previous work [42], in which the block was analyzed with empty air cavities.

The main findings of the work showed that the introduction of waste-insulating material has significantly improved the thermal performance of the honeycomb structure block, leading to a decrease in its transmittance of up to 56%. In particular, the sample containing sheep's wool resulted better than the other two, with a transmittance value equal to $0.53 \pm 0.02$ W/m$^2$K. However, the thermal images showed that the block with wood sawdust has a more homogeneous thermal behavior than the others due to the better distribution of the material inside the cavities.

The results obtained are promising, even if they are still far from the requirements requested for high-performance building envelopes. In any case, this study represents a step forward in the search for environmentally friendly solutions. Research in this direction can continue, testing new geometries to be created with 3D printing, other waste materials capable of guaranteeing thermal characteristics suitable for the creation of energy-efficient envelopes, and the search for printing materials with a low carbon footprint.

**Author Contributions:** Conceptualization, T.d.R.; methodology, T.d.R. and D.A.; investigation, T.d.R., A.C. and G.P.; resources, T.d.R. and A.C.; data curation, T.d.R., A.C. and G.P.; writing—original draft preparation, T.d.R. and A.C.; writing—review and editing, T.d.R., D.P. and D.A.; visualization, T.d.R., A.C. and G.P.; supervision, T.d.R., D.P. and D.A.; funding acquisition, T.d.R. All authors have read and agreed to the published version of the manuscript.

**Funding:** This research was partially funded by "Progetti di Ateneo per Avvio alla Ricerca, Decreto del Rettore n. 786 del 13.07.2021" of the University of L'Aquila, for the Project titled "BIT-3D".

**Institutional Review Board Statement:** Not applicable.

**Informed Consent Statement:** Not applicable.

**Data Availability Statement:** Not applicable.

**Acknowledgments:** T.d.R. thanks the University of L'Aquila for the financial support.

**Conflicts of Interest:** The authors declare no conflict of interest. The funders had no role in the design of the study; in the collection, analyses, or interpretation of data; in the writing of the manuscript; or in the decision to publish the results.

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
