# Peer review of "On the Use of Waste Materials for Thermal Improvement of 3D-Printed Block—An Experimental Comparison"

_buildings, doi:10.3390/buildings13051136_

Round 1

Reviewer 1 Report

The paper explores the use of waste materials for the thermal insulation of 3D-Printed blocks. The reviewer suggests that the authors address some comments prior to publication.

·       Overall, very good introduction. Well supported with references.

·       Introduction. Line 33-35. Please support this sentence with a reference.

·       In the introduction, the authors refer multiples time to the environmental impact of materials and processes. I would suggest referring to carbon footprint instead since this is a quantifiable metric many authors have used in previous research.

·       In the introduction and throughout the text, it is unclear how additive manufacturing can improve the sustainability of the building process. Is this by a topological optimization of the structure? Or by the ability to print with renewable materials in the mix design? Please elaborate on that.

·       Materials and methods. The authors do not mention why a honeycomb structure was printed. What was the criteria behind this? Why this pattern and not a different one? What is the infill pattern's effect on the block's thermal property? Are there any studies in such regard? Please elaborate.

·       Figure 1 has multiple typos. Please correct.

·       Why was PLA chosen as the material for the printing process? To my understanding, PLA has a relatively high carbon footprint. Can this material be reused? Why was this material chosen and not a different one? How does this material compare to the material in a façade or building envelope in terms of thermal properties? Please explain.

·       What were the criteria for selecting the amount of material (mass) used in each block? Why do the three samples have different infill material masses? How can you compare the thermal properties of the blocks if each one has a different insulating-material mass? Please elaborate.

·       Results. Line 263. The times must be wrong. If it was measured from 7:30 pm to 7:30 am, that is only 12h. I think that is a typing error.

·       Is PLA currently used to build building envelopes? The results are extrapolated to this application, but I am unsure of why that material was used.

Author Response

Dear Reviewer,

the paper "On the Use of Waste Materials for Thermal Improvement of 3D-Printed Block – An Experimental Comparison" - buildings-2300763 has been revised, following your helpful remarks and suggestions, for which I would like to thank you: they lead, I believe, to a much-improved paper.

The main changes are listed below and highlighted in the manuscript, following your hints.

****************************************************************************** 

The paper explores the use of waste materials for the thermal insulation of 3D-Printed blocks. The reviewer suggests that the authors address some comments prior to publication.

Overall, very good introduction. Well supported with references.

Authors: Thank you for your positive opinion about the work and for your comments, following which, we believe that the article is much improved.

Introduction. Line 33-35. Please support this sentence with a reference.

Authors: Thank you for your comment. A reference was added to support the sentence.

In the introduction, the authors refer multiples time to the environmental impact of materials and processes. I would suggest referring to carbon footprint instead since this is a quantifiable metric many authors have used in previous research.

Authors: Thank you for your useful suggestion. The revised version of the manuscript has been modified accordingly.

In the introduction and throughout the text, it is unclear how additive manufacturing can improve the sustainability of the building process. Is this by a topological optimization of the structure? Or by the ability to print with renewable materials in the mix design? Please elaborate on that.

Authors: Thank you for your comment that allows us to clarify how additive manufacturing can improve the sustainability of the building process. In our humble opinion, two of the main reasons why 3D printing is a promising technology to produce insulating blocks are:

1) the topological optimization of the structure, creating complex geometries suitable for mitigating heat transfer phenomena;

2) the possibility of using the 3D printed blocks as housing for waste insulating materials.

Clearly, the search for printing materials with a low carbon footprint, which is not a goal of our research, is also an opportunity for the development of additive manufacturing.

Following your comment, the revised version of the manuscript has been modified, highlighting why 3D printing can be a promising technology for insulating materials.

Materials and methods. The authors do not mention why a honeycomb structure was printed. What was the criteria behind this? Why this pattern and not a different one? What is the infill pattern's effect on the block's thermal property? Are there any studies in such regard? Please elaborate.

Authors: Thank you for your comment. In the revised manuscript, we tried to make explicit why the choice of the block with a honeycomb structure and what is the main objective of the work, which is to fill the air cavities of the block with waste materials and evaluate the thermal performance for the different cases.

In the literature there are a few similar studies, cited in our paper, focusing on air cavity filling of 3D-printed blocks.

Figure 1 has multiple typos. Please correct.

Authors: Thank you for your comment. Figure 1 has been modified.

Why was PLA chosen as the material for the printing process? To my understanding, PLA has a relatively high carbon footprint. Can this material be reused? Why was this material chosen and not a different one? How does this material compare to the material in a façade or building envelope in terms of thermal properties? Please explain.

Authors: Thank you for your comment. The PLA was chosen first because it is one of the most widely used materials for 3D printing. Secondly, PLA is a recyclable and compostable bioplastic with low greenhouse gas emissions [Ghomi et al., The Life Cycle Assessment for Polylactic Acid (PLA) to Make It a Low-Carbon Material, Polymers 2021, 13(11), 1854]. A clarification was added in Section 1 of the revised paper.

What were the criteria for selecting the amount of material (mass) used in each block? Why do the three samples have different infill material masses? How can you compare the thermal properties of the blocks if each one has a different insulating-material mass? Please elaborate.

Authors: Thank you for your comment that allows us to detail how the blocks were prepared.

The amount of material in the air cavities of the blocks varies due to the different densities and types of materials chosen. Wood sawdust, which is very fine, allows more material to fit into the cavities than other materials. In contrast, sheep wool and hemp are used as filaments. Therefore, the material masses vary.

Following your comment, more information has been added in section 2.3 of the revised manuscript.

Results. Line 263. The times must be wrong. If it was measured from 7:30 pm to 7:30 am, that is only 12h. I think that is a typing error.

Authors: Thank you for your comment. The times of the experimental campaigns have been corrected.

Is PLA currently used to build building envelopes? The results are extrapolated to this application, but I am unsure of why that material was used.

Authors: Thank you for your comment.

To date, PLA is not used for building envelopes, as well as 3D printing in general, except for sporadic examples. However, in the humble opinion of the authors, the 3D printing and the results obtained in our works show that this could be a promising technology for future application in the building sector. Clearly, the authors agree with the Reviewer’s comment: the performance obtained is far from the levels required for the building sector today, as made explicit in the text. (See line 342 of the Conclusions).

Reviewer 2 Report

Dear editor

Here are my opinions and suggestions for the article titled " On the Use of Waste Materials for Thermal Improvement of 3D-Printed Block – An Experimental Analysis", to that I was assigned as a referee.

This work aims to expand a previous study conducted by Rubeis et al.  to understand how the use of waste materials can improve the thermal behavior of the honeycomb-structure 3D printed block. So reader can easily understand why honeycomb-structure is selected.

The article is well constructed, the method is well explained, and the hypotheses are reliably tested. I think the article is suitable for publication. As a small suggestion, the reader could realize that the authors made a comparison study by using different materials namely, Wood sawdust, Sheep's wool and Hemp. However, the reader cannot understand in the title and introduction that the authors are doing a comparison study. However, (s)he sees it in the Method section. It may be appropriate to include this important information in the introduction and even to include it in the title. Furthermore, the selection criteria for these three materials can be explained. Why authors selected those 3? Are there any other options? If so, why they were skipped? In this respect, the “research question and hypothesis” can be revised accordingly.

Author Response

Dear Reviewer,

the paper "On the Use of Waste Materials for Thermal Improvement of 3D-Printed Block – An Experimental Comparison" - buildings-2300763 has been revised, following your helpful remarks and suggestions, for which I would like to thank you: they lead, I believe, to a much-improved paper.

The main changes are listed below and highlighted in the manuscript, following your hints.

*************************************************************************************

Dear editor

Here are my opinions and suggestions for the article titled " On the Use of Waste Materials for Thermal Improvement of 3D-Printed Block – An Experimental Analysis", to that I was assigned as a referee.

This work aims to expand a previous study conducted by Rubeis et al.  to understand how the use of waste materials can improve the thermal behavior of the honeycomb-structure 3D printed block. So reader can easily understand why honeycomb-structure is selected.

The article is well constructed, the method is well explained, and the hypotheses are reliably tested. I think the article is suitable for publication. As a small suggestion, the reader could realize that the authors made a comparison study by using different materials namely, Wood sawdust, Sheep's wool and Hemp. However, the reader cannot understand in the title and introduction that the authors are doing a comparison study. However, (s)he sees it in the Method section. It may be appropriate to include this important information in the introduction and even to include it in the title.

Authors: Thank you for your positive opinion about the work and for your comments, following which, we believe that the article is much improved.

Following your suggestion, the title, abstract, and introduction have been modified highlighting that the work presents a comparison between different waste insulating materials.

Furthermore, the selection criteria for these three materials can be explained. Why authors selected those 3? Are there any other options? If so, why they were skipped? In this respect, the “research question and hypothesis” can be revised accordingly.

Authors: Thank you for your comment. The choice of materials is based on their characteristics (natural and waste) and whether they can be used to fill the air cavities of the 3D-printed blocks.

Following your comment, a clarification has been added in section 2 of the revised manuscript.

Reviewer 3 Report

After analysis of the article, "On the Use of Waste Materials for Thermal Improvement of 3D-2 Printed Block – An Experimental Analysis" I have the following observations:

The authors notes that the insulation materials were chosen "After a thorough literature search…" (line 182). It would be useful to know more about the analysis and selection criteria. Thermal conductivity of all selected materials varies between 0.038 and 0.060 W/mK?  Could a similar interval have influenced the results? Authors should describe this point more detail. Also, it is necessary to give the main properties of insulating materials, selected for investigation.

The discussion of the obtained results should have more scientific insights. As known, thermal resistance of insulating materials depends on density of a material. My recommendation is to give the considerations on relationship between materials density and obtained results. In addition, the characteristics of samples should fulfilled (not only mass/dimensions). How was the amount of material needed to fill the blocks determined?

In 2.4. section I  found identical information as published in authors’ previous paper de Rubeis, T.; Ciccozzi, A.; Giusti, L.; Ambrosini, D. The 3D printing potential for heat flow optimization - Influence of block 419 geometries on heat transfer processes. Sustainability 2022, 14(23), 15830. This information have to be corrected giving the reference to previous paper.

Author Response

Dear Reviewer,

the paper "On the Use of Waste Materials for Thermal Improvement of 3D-Printed Block – An Experimental Comparison" - buildings-2300763 has been revised, following your helpful remarks and suggestions, for which I would like to thank you: they lead, I believe, to a much-improved paper.

The main changes are listed below and highlighted in the manuscript, following your hints.

*************************************************************************************

After analysis of the article, "On the Use of Waste Materials for Thermal Improvement of 3D-2 Printed Block – An Experimental Analysis" I have the following observations:

The authors notes that the insulation materials were chosen "After a thorough literature search…" (line 182). It would be useful to know more about the analysis and selection criteria.

Authors: Thank you for your comment. More information has been added to explain how the selection of materials took place.

Thermal conductivity of all selected materials varies between 0.038 and 0.060 W/mK?  Could a similar interval have influenced the results? Authors should describe this point more detail. Also, it is necessary to give the main properties of insulating materials, selected for investigation.

Authors: Thank you for your comment. Certainly, variations in thermal conductivity led to variations in the results - in terms of thermal conductance and thermal transmittance. Moreover, the presence of air in the insulating materials, which depends on the density of the material, also determines the effects on the experimental results.

Following your comment, more details have been included in section 2.3.

The discussion of the obtained results should have more scientific insights. As known, the thermal resistance of insulating materials depends on the density of a material. My recommendation is to give the consideration on the relationship between materials density and obtained results. In addition, the characteristics of samples should fulfilled (not only mass/dimensions). How was the amount of material needed to fill the blocks determined?

Authors: Thank you for your comment. The thermal resistance of insulating materials is obtained by the ratio between the thickness (s) of the material and its thermal conductivity (λ), i.e., following the equation:

Rcond=s/λ  [m2K/W]

However, the experimental analysis was performed through the HFM method, which determines the thermal conductance of the block based on measurements of heat flux and indoor and outdoor surface temperatures. The choice of using an experimental approach (HFM method) is due to the idea of not relying on the thermal conductivity of the materials.

Anyway, following your comment, the characteristics of the samples (amount of materials, density) were added in the revised version of the manuscript.

In 2.4. section I  found identical information as published in authors’ previous paper de Rubeis, T.; Ciccozzi, A.; Giusti, L.; Ambrosini, D. The 3D printing potential for heat flow optimization - Influence of block 419 geometries on heat transfer processes. Sustainability 2022, 14(23), 15830. This information has to be corrected giving the reference to previous paper.

Authors: Thank you for your comment. The reference to the previous paper was added in section 2.4.

Round 2

Reviewer 1 Report

Dear Authors,

I think the manuscript has been significantly improved after your revision. I am pretty satisfied with the answers to all my comments, and I do not have any further comments. I suggest the paper is published as is.

Reviewer 3 Report

I think the article “On the Use of Waste Materials for Thermal Improvement of 3D-2 Printed Block – An Experimental Comparison” has been corrected mostly following the given comments. No deeper analysis of the results was performed, as I understand. In my opinion, in the results section more widely presented insights could increase the scientific value of the article.  However, this can not be an obstacle to publicizing the article.